# MESA: Boost Ensemble Imbalanced Learning with MEta-SAmpler

**Zhining Liu**
Jilin University
znliu19@mails.jlu.edu.cn

**Pengfei Wei**
National University of Singapore
dcsweip@nus.edu.sg

**Jing Jiang**
University of Technology Sydney
jing.jiang@uts.edu.au

**Wei Cao**
Microsoft Research
weicao@microsoft.com

**Jiang Bian**
Microsoft Research
jiang.bian@microsoft.com

**Yi Chang**[*]
Jilin University
yichang@jlu.edu.cn

## Abstract

*Imbalanced learning* (IL), i.e., learning unbiased models from class-imbalanced data, is a challenging problem. Typical IL methods including resampling and reweighting were designed based on some heuristic assumptions. They often suffer from unstable performance, poor applicability, and high computational cost in complex tasks where their assumptions do not hold. In this paper, we introduce a novel ensemble IL framework named MESA. It adaptively resamples the training set in iterations to get multiple classifiers and forms a cascade ensemble model. MESA directly learns the sampling strategy from data to optimize the final metric beyond following random heuristics. Moreover, unlike prevailing meta-learning-based IL solutions, we decouple the model-training and meta-training in MESA by independently train the meta-sampler over task-agnostic meta-data. This makes MESA generally applicable to most of the existing learning models and the meta-sampler can be efficiently applied to new tasks. Extensive experiments on both synthetic and real-world tasks demonstrate the effectiveness, robustness, and transferability of MESA. Our code is available at https://github.com/ZhiningLiu1998/mesa.

## 1  Introduction

*Class imbalance*, due to the naturally-skewed class distributions, has been widely observed in many real-world applications such as click prediction, fraud detection, and medical diagnosis [13, 15, 20]. Canonical classification algorithms usually induce the bias, i.e., perform well in terms of global accuracy but poorly on the minority class, in solving class imbalance problems. However, the minority class commonly yields higher interests from both learning and practical perspectives [18, 19].

Typical imbalanced learning (IL) algorithms attempt to eliminate the bias through data *resampling* [6, 16, 17, 25, 32] or *reweighting* [27, 30, 36] in the learning process. More recently, ensemble learning is incorporated to reduce the variance introduced by resampling or reweighting and has achieved satisfactory performance [22]. In practice, however, all these methods have been observed to suffer from three major limitations: (I) unstable performance due to the sensitivity to outliers, (II)

---

[*]Corresponding author.

poor applicability because of the prerequisite of domain experts to hand-craft the cost matrix, and (III) high cost of computing the distance between instances.

Regardless the computational issue, we attribute the unsatisfactory performance of traditional IL methods to the validity of heuristic assumptions made on training data. For instance, some methods [7, 12, 29, 35] assume instances with higher training errors are more informative for learning. However, misclassification may be caused by outliers, and error reinforcement arises in this case with the above assumption. Another widely used assumption is that generating synthetic samples around minority instances helps with learning [7, 8, 42]. This assumption only holds when the minority data is well clustered and sufficiently discriminative. If the training data is extremely imbalanced or with many corrupted labels, the minority class would be poorly represented and lack a clear structure. In this case, working under this assumption severely jeopardizes the performance.

Henceforth, it is much more desired to develop an adaptive IL framework that is capable of handling complex real-world tasks without intuitive assumptions. Inspired by the recent developments in meta-learning [24], we propose to achieve the meta-learning mechanism in ensemble imbalanced learning (EIL) framework. In fact, some preliminary efforts [33, 34, 37] have investigated the potential of applying meta-learning to IL problems. Nonetheless, these works have limited capability of generalization because of the model-dependent optimization process. Their meta-learners are confined to be co-optimized with a single DNN, which greatly limits their application to other learning models (e.g., tree-based models) as well as deployment into the more powerful EIL framework.

In this paper, we propose a generic EIL framework MESA that automatically learns its strategy, i.e., the meta-sampler, from data towards optimizing imbalanced classification. The main idea is to model a meta-sampler that serves as an adaptive under-sampling solution embedded in the iterative ensemble training process. In each iteration, it takes the current state of ensemble training (i.e., the classification error distribution on both the training and validation sets) as its input. Based on this, the meta-sampler selects a subset to train a new base classifier and then adds it to the ensemble, a new state can thus be obtained. We expect the meta-sampler to maximize the final generalization performance by learning from such interactions. To this end, we use reinforcement learning (RL) to solve the non-differentiable optimization problem of the meta-sampler. To summarize, this paper makes the following contributions. (I) We propose MESA, a generic EIL framework that demonstrates superior performance by automatically learning an adaptive under-sampling strategy from data. (II) We carry out a preliminary exploration of extracting and using cross-task meta-information in EIL systems. The usage of such meta-information gives the meta-sampler cross-task transferability. A pretrained meta-sampler can be directly applied to new tasks, thereby greatly reducing the computational cost brought about by meta-training. (III) Unlike prevailing methods whose meta-learners were designed to be co-optimized with a specific learning model (i.e, DNN) during training, we decoupled the model-training and meta-training process in MESA. This makes our framework generally applicable to most of the statistical and non-statistical learning models (e.g., decision tree, Naïve Bayes, k-nearest neighbor classifier).

## 2 Related Work

Fernández et al. [1], Guo et al. [15], and He et al. [18, 19] provided systematic reviews of algorithms and applications of imbalanced learning. In this paper, we focus on *binary imbalanced classification* problem, which is one of the most widely studied problem setting [15, 22] in imbalanced learning. Such a problem extensively exists in practical applications, e.g., fraud detection (fraud vs. normal), medical diagnosis (sick vs. healthy), and cybersecurity (intrusion vs. user connection). We mainly review existing works on this problem as follows.

**Resampling** Resampling methods focus on modifying the training set to balance the class distribution (i.e., over/under-sampling [6, 16, 17, 32, 38]) or filter noise (i.e., cleaning resampling [25, 41]). Random resampling usually leads to severe information loss or overfishing, hence many advanced methods explore distance information to guide their sampling process [15]. However, calculating the distance between instances is computationally expensive on large-scale datasets, and such strategies may even fail to work when the data does not fit their assumptions.

**Reweighting** Reweighting methods assign different weights to different instances to alleviate a classifier's bias towards majority groups (e.g., [5, 12, 28, 30]). Many recent reweighting methods such as FocalLoss [27] and GHM [26] are specifically designed for DNN loss function engineering.

Table 1: Comparisons of MESA with existing imbalanced learning methods, note that $|\mathcal{N}| \gg |\mathcal{P}|$.

| Category* | Representative(s) | Sample efficiency | Distance-based resampling cost | Domain knowledge free? | Robust to noises/outliers? | Requirements |
|---|---|---|---|---|---|---|
| RW | [28], [5] | $\mathcal{O}(|\mathcal{P}| + |\mathcal{N}|)$ | ✗ | ✗ | ✗ | cost matrix set by domain experts |
| US | [32], [38] | $\mathcal{O}(2|\mathcal{P}|)$ | $\mathcal{O}(|\mathcal{P}|)$ | ✓ | ✗ | well-defined distance metric |
| OS | [6], [17] | $\mathcal{O}(2|\mathcal{N}|)$ | $\mathcal{O}(|\mathcal{P}|)$ | ✓ | ✗ | well-defined distance metric |
| CS | [43], [40] | $\mathcal{O}(|\mathcal{P}| + |\mathcal{N}|)$ | $\mathcal{O}(|\mathcal{P}| \cdot |\mathcal{N}|)$ | ✓ | ✓ | well-defined distance metric |
| OS+CS | [4], [3] | $\mathcal{O}(2|\mathcal{N}|)$ | $\mathcal{O}(|\mathcal{P}| \cdot |\mathcal{N}|)$ | ✓ | ✓ | well-defined distance metric |
| IE+RW | [12], [39] | $\mathcal{O}(k(|\mathcal{P}| + |\mathcal{N}|))$ | ✗ | ✗ | ✗ | cost matrix set by domain experts |
| PE+US | [2], [29] | $\mathcal{O}(2k|\mathcal{P}|)$ | ✗ | ✓ | ✓ | - |
| PE+OS | [42] | $\mathcal{O}(2k|\mathcal{N}|)$ | $\mathcal{O}(2k|\mathcal{P}|)$ | ✓ | ✓ | well-defined distance metric |
| IE+RW+US | [35] | $\mathcal{O}(2k|\mathcal{P}|)$ | ✗ | ✗ | ✗ | - |
| IE+RW+OS | [7] | $\mathcal{O}(2k|\mathcal{N}|)$ | $\mathcal{O}(2k|\mathcal{P}|)$ | ✓ | ✗ | well-defined distance metric |
| ML | [37], [34], [44] | $\mathcal{O}(|\mathcal{P}| + |\mathcal{N}|)$ | ✗ | ✗ | ✓ | co-optimized with DNN only |
| IE+ML | MESA(ours) | $\mathcal{O}(2k|\mathcal{P}|)$ | ✗ | ✓ | ✓ | independent meta-training |

* reweighting (RW), under-sampling (US), over-sampling (OS), cleaning-sampling (CS), iterative ensemble (IE), parallel ensemble (PE), meta-learning (ML).

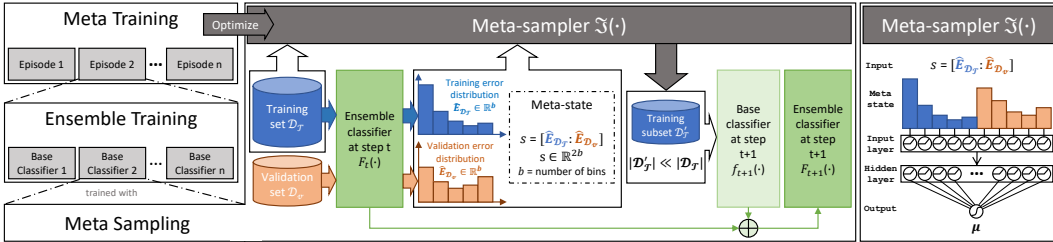

Figure 1: Overview of the proposed MESA Framework. Best viewed in color.

Class-level reweighting such as cost-sensitive learning [30] is more versatile but requires a cost matrix given by domain experts beforehand, which is usually infeasible in practice.

**Ensemble Methods.** Ensemble imbalanced learning (EIL) is known to effectively improve typical IL solutions by combining the outputs of multiple classifiers (e.g., [7, 29, 31, 35, 42]). These EIL approaches prove to be highly competitive [22] and thus gain increasing popularity [15] in IL. However, most of the them are straight combinations of a resampling/reweighting solution and an ensemble learning framework, e.g., SMOTE [6]+ADABOOST [12]=SMOTEBOOST [7]. Consequently, although EIL techniques effectively lower the variance introduced by resampling/reweighting, these methods still suffer from unsatisfactory performance due to their heuristic-based designs.

**Meta-learning Methods.** Inspired by recent meta-learning developments [11, 24], there are some studies that adapt meta-learning to solve IL problem. Typical methods include Learning to Teach [44] that learns a dynamic loss function, MentorNet [21] that learns a mini-batch curriculum, and L2RW [34]/Meta-Weight-Net [37] that learn an implicit/explicit data weighting function. Nonetheless, all these methods are confined to be co-optimized with a DNN by gradient descent. As the success of deep learning relies on the massive training data, mainly from domains like computer vision and natural language processing, the applications of these methods to other learning models (e.g., tree-based models and their ensemble variants like gradient boosting machine) in traditional classification tasks (e.g., small/structured/tabular data) are highly constrained.

We present a comprehensive comparison of existing IL solutions for binary imbalanced classification problem with our MESA in Table 1. Compared with other methods, MESA aims to learn a resampling strategy directly from data. It is able to perform quick and adaptive resampling as no distance computing, domain knowledge, or related heuristics are involved in the resampling process.

## 3 The proposed MESA framework

In order to take advantage of both ensemble learning and meta-learning, we propose a novel EIL framework named MESA that works with a meta-sampler. As shown in Fig. 1, MESA consists of three parts: *meta-sampling* as well as *ensemble training* to build ensemble classifiers, and *meta-training* to optimize the meta-sampler. We will describe them respectively in this section.

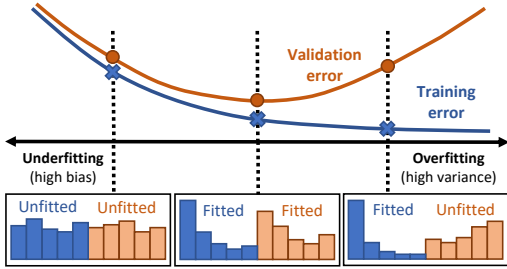

Figure 2: Some examples of different meta-states ($s = [\widehat{E}_{\mathcal{D}_\tau} : \widehat{E}_{\mathcal{D}_v}]$) and their corresponding ensemble training states. The meta-state reflects how well the current classifier fits on the training set, and how well it generalizes to unseen validation data. Note that such representation is independent of properties of the specific task (e.g., dataset size, feature space) thus can be used to support the meta-sampler to perform adaptive resampling across different tasks.

Specifically, MESA is designed to: (I) perform adaptive resampling based on meta-information to further boost the performance of ensemble classifiers; (II) decouple model-training and meta-training for general applicability to different classifiers; (III) train the meta-sampler over task-agnostic meta-data for cross-task transferability and reducing meta-training cost on new tasks.

**Notations.** Let $\mathcal{X} : \mathbb{R}^d$ be the input feature space and $\mathcal{Y} : \{0, 1\}$ be the label space. An instance is represented by $(x, y)$, where $x \in \mathcal{X}$, $y \in \mathcal{Y}$. Without loss of generality, we always assume that the minority class is positive. Given an imbalanced dataset $\mathcal{D} : \{(x_1, y_1), (x_2, y_2), \cdots, (x_n, y_n)\}$, the minority set is $\mathcal{P} : \{(x, y) \mid y = 1, (x, y) \in \mathcal{D}\}$ and the majority set is $\mathcal{N} : \{(x, y) \mid y = 0, (x, y) \in \mathcal{D}\}$. For highly imbalanced data we have $|\mathcal{N}| \gg |\mathcal{P}|$. We use $f : x \to [0, 1]$ to denote a single classifier and $F_k : x \to [0, 1]$ to denote an ensemble classifier that is formed by $k$ base classifiers. We use $\mathcal{D}_\tau$ and $\mathcal{D}_v$ to represent the training set and validation set, respectively.

**Meta-state.** As mentioned before, we expect to find a task-agnostic representation that can provide the meta-sampler with the information of the ensemble training process. Motivated by the concept of "gradient/hardness distribution" from [26, 31], we introduce the histogram distribution of the training and validation errors as the meta-state of the ensemble training system.

Formally, given an data instance $(x, y)$ and an ensemble classifier $F_t(\cdot)$, the classification error $e$ is defined as the absolute difference between the predicted probability of $x$ being positive and the ground truth label $y$, i.e., $|F_t(x) - y|$. Suppose the error distribution on dataset $\mathcal{D}$ is $E_\mathcal{D}$, then the error distribution approximated by histogram is given by a vector $\widehat{E}_\mathcal{D} \in \mathbb{R}^b$, where $b$ is the number of bins in the histogram. Specifically, the $i$-th component of vector $\widehat{E}_\mathcal{D}$ can be computed as follows[2]:

$$\widehat{E}_\mathcal{D}^i = \frac{|\{(x, y) \mid \frac{i-1}{b} \leq abs(F_t(x) - y) < \frac{i}{b}, (x, y) \in \mathcal{D}\}|}{|\mathcal{D}|}, 1 \leq i \leq b. \tag{1}$$

After concatenating the error distribution vectors on training and validation set, we have the meta-state:

$$s = [\widehat{E}_{\mathcal{D}_\tau} : \widehat{E}_{\mathcal{D}_v}] \in \mathbb{R}^{2b}. \tag{2}$$

Intuitively, the histogram error distribution $\widehat{E}_\mathcal{D}$ shows how well the given classifier fits the dataset $\mathcal{D}$. When $b = 2$, it reports the accuracy score in $\widehat{E}_\mathcal{D}^1$ and misclassification rate in $\widehat{E}_\mathcal{D}^2$ (classification threshold is 0.5). With $b > 2$, it shows the distribution of "easy" samples (with errors close to 0) and "hard" samples (with errors close to 1) in finer granularity, thus contains more information to guide the resampling process. Moreover, since we consider both the training and validation set, the meta-state also provides the meta-sampler with information about bias/variance of the current ensemble model and thus supporting its decision. We show some illustrative examples in Fig. 2.

**Meta Sampling.** Making instance-level decisions by using a complex meta-sampler (e.g., set a large output layer or use recurrent neural network) is extremely time-consuming as the complexity of a single update $C_u$ is $\mathcal{O}(|\mathcal{D}|)$. Besides, complex model architecture also brings extra memory cost and hardship in optimization. To make MESA more concise and efficient, we use a Gaussian function trick to simplify the meta-sampling process and the sampler itself, reducing $C_u$ from $\mathcal{O}(|\mathcal{D}|)$ to $\mathcal{O}(1)$.

Specifically, let $\Im$ denote the meta-sampler, it outputs a scalar $\mu \in [0, 1]$ based on the input meta-state $s$, i.e., $\mu \sim \Im(\mu|s)$. We then apply a Gaussian function $g_{\mu,\sigma}(x)$ over each instance's classification

**Algorithm 1** $\mathtt{Sample}(\mathcal{D}_\tau; F, \mu, \sigma)$

**Require:** $\mathcal{D}_\tau, F, \mu, \sigma$
1: Initialization: derive minority set $\mathcal{P}_\tau$ and majority set $\mathcal{N}_\tau$ from $\mathcal{D}_\tau$
2: Assign each $(x_i, y_i)$ in $\mathcal{N}_\tau$ with weight:

$$w_i = \frac{g_{\mu,\sigma}(|F(x_i) - y_i|)}{\sum_{(x_j, y_j) \in \mathcal{N}_\tau} g_{\mu,\sigma}(|F(x_j) - y_j|)}$$

3: Sample majority subset $\mathcal{N}_\tau'$ from $\mathcal{N}_\tau$ w.r.t. sampling weights $w$, where $|\mathcal{N}_\tau'| = |\mathcal{P}_\tau|$
4: **return** balanced subset $\mathcal{D}_\tau' = \mathcal{N}_\tau' \cup \mathcal{P}_\tau$

**Algorithm 2** Ensemble training in MESA

**Require:** $\mathcal{D}_\tau, \mathcal{D}_v, \Im, \sigma, f, b, k$
1: train $f_1(x)$ with random balanced subset
2: **for** $t$=1 to $k-1$ **do**
3:     $F_t(x) = \frac{1}{t} \sum_{i=1}^{t} f_i(x)$
4:     compute $\widehat{E}_{\mathcal{D}_\tau}$ and $\widehat{E}_{\mathcal{D}_v}$ by Eq. 1
5:     $s_t = [\widehat{E}_{\mathcal{D}_\tau} : \widehat{E}_{\mathcal{D}_v}]$
6:     $\mu_t \sim \Im(\mu_t | s_t)$
7:     $\mathcal{D}_{t+1,\tau}' = \mathtt{Sample}(\mathcal{D}_\tau; F_t, \mu_t, \sigma)$
8:     train new classifier $f_{t+1}(x)$ with $\mathcal{D}_{t+1,\tau}'$
9: **return** $F_k(x) = \frac{1}{k} \sum_{i=1}^{k} f_i(x)$

**Algorithm 3** Meta-training in MESA

1: Initialization: replay memory $\mathcal{M}$ with capacity $N$, network parameters $\psi, \bar{\psi}, \theta,$ and $\varphi$
2: **for** episode = 1 to $M$ **do**
3:     **for** each environment step $t$ **do**
4:         observe $s_t$ from ENV         ▷ line3-5 in Alg. 2
5:         take action $\mu_t \sim \Im_\varphi(\mu_t | s_t)$         ▷ line6-8 in Alg. 2
6:         observe reward $r_t = P(F_{t+1}, \mathcal{D}_v) - P(F_t, \mathcal{D}_v)$ and $s_{t+1}$
7:         store transition $\mathcal{M} = \mathcal{M} \cup \{(s_t, \mu_t, r_t, s_{t+1})\}$
8:     **for** each gradient step **do**
9:         update $\psi, \bar{\psi}, \theta,$ and $\varphi$ according to [14]
10: **return** meta-sampler $\Im$ with parameters $\varphi$

error to decide its (unnormalized) sampling weight, where $g_{\mu,\sigma}(x)$ is defined as:

$$g_{\mu,\sigma}(x) = \frac{1}{\sigma\sqrt{2\pi}} e^{-\frac{1}{2}(\frac{x-\mu}{\sigma})^2}. \tag{3}$$

Note that in Eq. 3, $e$ is the Euler's number, $\mu \in [0, 1]$ is given by the meta-sampler and $\sigma$ is a hyper-parameter. Please refer to the Appendix for discussions and guidelines about our hyper-parameter setting. The above meta-sampling procedure $\mathtt{Sample}(\ \cdot\ ; F, \mu, \sigma)$ is summarized in Algorithm 1.

**Ensemble Training.** Given a meta-sampler $\Im : \mathbb{R}^{2b} \to [0, 1]$ and the meta-sampling strategy, we can iteratively train new base classifiers using the dataset sampled by the sampler. At the $t$-th iteration, having the current ensemble $F_t(\cdot)$, we can obtain $\widehat{E}_{\mathcal{D}_\tau}$, $\widehat{E}_{\mathcal{D}_v}$ and meta-state $s_t$ by applying Eqs. (1) and (2). Then a new base classifier $f_{t+1}(\cdot)$ is trained with the subset $\mathcal{D}_{t+1,\tau}' = \mathtt{Sample}(\mathcal{D}_\tau; F_t, \mu_t, \sigma)$, where $\mu_t \sim \Im(\mu_t | s_t)$ and $\mathcal{D}_\tau$ is the original training set. Note that $f_1(\cdot)$ was trained on a random balanced subset, as there is no trained classifier in the first iteration. See Algorithm 2 for more details.

**Meta Training.** As described above, our meta-sampler $\Im$ is trained to optimize the generalized performance of an ensemble classifier by iteratively selecting its training data. It takes the current state $s$ of the training system as input, and then outputs the parameter $\mu$ of a Gaussian function to decide each instance's sampling probability. The meta-sampler is expected to learn and adapt its strategy from such state($s$)-action($\mu$)-state(new $s$) interactions. The non-differentiable optimization problem of training $\Im$ can thus be naturally approached via reinforcement learning (RL).

We consider the ensemble training system as the environment (ENV) in the RL setting. The corresponding Markov decision process (MDP) is defined by the tuple $(\mathcal{S}, \mathcal{A}, p, r)$, where the state space $\mathcal{S} : \mathbb{R}^{2b}$ and action space $\mathcal{A} : [0, 1]$ is continuous, and the unknown state transition probability $p : \mathcal{S} \times \mathcal{S} \times \mathcal{A} \to [0, \infty)$ represents the probability density of the next state $s_{t+1} \in \mathcal{S}$ given the current state $s_t \in \mathcal{S}$ and action $a_t \in \mathcal{A}$. More specifically, in each episode, we iteratively train $k$ base classifiers $f(\cdot)$ and form a cascade ensemble classifier $F_k(\cdot)$. In each environment step, ENV provides the meta-state $s_t = [\widehat{E}_{\mathcal{D}_\tau} : \widehat{E}_{\mathcal{D}_v}]$, and then the action $a_t$ is selected by $a_t \sim \Im(\mu_t | s_t)$, i.e., $a_t \Leftrightarrow \mu_t$. A new base classifier $f_{t+1}(\cdot)$ is trained using the subset $\mathcal{D}_{t+1,\tau}' = Sample(\mathcal{D}_\tau; F_t, a_t, \sigma)$.

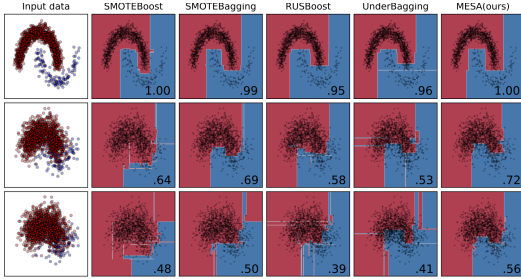

Figure 3: Comparisons of MESA with 4 representative traditional EIL methods (SMOTE-BOOST [7], SMOTEBAGGING [42], RUSBOOST [35] and UNDERBAGGING [2]) on 3 toy datasets with different levels of underlying class distribution overlapping (less/mid/highly-overlapped in 1st/2nd/3rd row). The number in the lower right corner of each subfigure represents the AUCPRC score of the corresponding classifier. Best viewed in color.

After adding $f_{t+1}(\cdot)$ into the ensemble, the new state $s_{t+1}$ was sampled w.r.t. $s_{t+1} \sim p(s_{t+1}; s_t, a_t)$. Given a performance metric function $P(F, \mathcal{D}) \rightarrow \mathbb{R}$, the reward $r$ is set to the generalization performance difference of $F$ before and after an update (using the keep-out validation set for unbiased estimation), i.e., $r_t = P(F_{t+1}, \mathcal{D}_v) - P(F_t, \mathcal{D}_v)$. The optimization goal of the meta-sampler (i.e., the cumulative reward) is thus the generalization performance of the ensemble classifier.

We take advantage of Soft Actor-Critic [14] (SAC), an off-policy actor-critic deep RL algorithm based on the maximum entropy RL framework, to optimize our meta-sampler $\Im$. In our case, we consider a parameterized state value function $V_\psi(s_t)$ and its corresponding target network $V_{\bar\psi}(s_t)$, a soft Q-function $Q_\theta(s_t, a_t)$, and a tractable policy (meta-sampler) $\Im_\varphi(a_t|s_t)$. The parameters of these networks are $\psi, \bar\psi, \theta,$ and $\varphi$. The rules for updating these parameters are given in the SAC paper [14]. We summarize the meta-training process of $\Im_\varphi$ in Algorithm 3.

**Complexity analysis.** Please refer to the Appendix (provided in supplementary material) for detailed complexity analysis of MESA alongside with related validating experiments.

## 4 Experiments

To thoroughly assess the effectiveness of MESA, two series of experiments are conducted: one on controlled synthetic toy datasets for visualization and the other on real-world imbalanced datasets to validate MESA's performance in practical applications. We also carry out extended experiments on real-world datasets to verify the robustness and cross-task transferability of MESA.

### 4.1 Experiment on Synthetic Datasets

**Setup Details.** We build a series of imbalanced toy datasets corresponding to different levels of underlying class distribution overlapping, as shown in Fig. 3. All the datasets have the same imbalance ratio[3] ($|\mathcal{N}|/|\mathcal{P}| = 2,000/200 = 10$). In this experiment, MESA is compared with four representative EIL algorithms from 4 major EIL branches (Parallel/Iterative Ensemble + Under/Over-sampling), i.e., SMOTEBOOST [7], SMOTEBAGGING [42], RUSBOOST [35], and UNDERBAGGING [2]. All EIL methods are deployed with decision trees as base classifiers with ensemble size of 5.

**Visualization & Analysis.** We plot the input datasets and the decision boundaries learned by different EIL algorithms in Fig. 3, which shows that MESA achieves the best performance under different situations. We can observe that: all tested methods perform well on the less-overlapped dataset (1st row). Note that random under-sampling discards some important majority samples (e.g., data points at the right end of the "∩"-shaped distribution) and cause information loss. This makes the performance of RUSBOOST and UNDERBAGGING slightly weaker than their competitors. As overlapping intensifies (2nd row), an increasing amount of noise gains high sample weights during the training process of boosting-based methods, i.e., SMOTEBOOST and RUSBOOST, thus resulting in poor classification performance. Bagging-based methods, i.e., SMOTEBAGGING and UNDERBAGGING, are less influenced by noise but they still underperform MESA. Even on the extremely overlapped dataset (3rd row), MESA still gives a stable and reasonable decision boundary that fits the underlying distribution. All the results show the superiority of MESA to other traditional EIL baselines in handling the overlapping, noises, and poor minority class representation.

Table 2: Comparisons of MESA with other representative resampling methods.

| Category | Method | Protein Homo. (IR=111) | | | | | #Training | Resampling |
| | | KNN | GNB | DT | Boost | GBM | Samples | Time (s) |
| --- | --- | --- | --- | --- | --- | --- | --- | --- |
| No resampling | - | 0.466 | 0.742 | 0.531 | 0.778 | 0.796 | 87,450 | - |
| Under-sampling | RANDOMUS | 0.146 | 0.738 | 0.071 | 0.698 | 0.756 | 1,554 | 0.068 |
| | NEARMISS [32] | 0.009 | 0.012 | 0.012 | 0.400 | 0.266 | 1,554 | 3.949 |
| Cleaning-sampling | CLEAN [25] | 0.469 | 0.744 | 0.488 | 0.781 | 0.811 | 86,196 | 117.739 |
| | ENN [43] | 0.460 | 0.744 | 0.532 | 0.789 | 0.817 | 86,770 | 120.046 |
| | TOMEKLINK [41] | 0.466 | 0.743 | 0.524 | 0.778 | 0.791 | 87,368 | 90.633 |
| | ALLKNN [40] | 0.459 | 0.744 | 0.542 | 0.789 | 0.816 | 86,725 | 327.110 |
| | OSS [23] | 0.466 | 0.743 | 0.536 | 0.778 | 0.789 | 87,146 | 92.234 |
| Over-sampling | RANDOMOS | 0.335 | 0.706 | 0.505 | 0.736 | 0.733 | 173,346 | 0.098 |
| | SMOTE [6] | 0.189 | 0.753 | 0.304 | 0.700 | 0.719 | 173,346 | 0.576 |
| | ADASYN [17] | 0.171 | 0.679 | 0.315 | 0.717 | 0.693 | 173,366 | 2.855 |
| | BORDERSMOTE [16] | 0.327 | 0.743 | 0.448 | 0.795 | 0.711 | 173,346 | 2.751 |
| Over-sampling + Cleaning | SMOTEENN [4] | 0.156 | 0.750 | 0.308 | 0.711 | 0.750 | 169,797 | 156.641 |
| | SMOTETOMEK [3] | 0.185 | 0.749 | 0.292 | 0.782 | 0.703 | 173,346 | 116.401 |
| Meta-sampler | MESA (OURS, $k$=10) | **0.585** | **0.804** | **0.832** | **0.849** | **0.855** | *1,554×10* | *0.235×10* |

## 4.2 Experiment on Real-world Datasets

**Setup Details.** In order to verify the effectiveness of MESA in practical applications, we extend the experiments to real-world imbalanced classification tasks from the UCI repository [10] and KDD CUP 2004. To ensure a thorough assessment, these datasets vary widely in their properties, with the imbalance ratio (IR) ranging from 9.1:1 to 111:1, dataset sizes ranging from 531 to 145,751, and number of features ranging from 6 to 617, please see Appendix provided in the supplementary material for detailed information. For each dataset, we keep-out the 20% validation set and report the result of 4-fold stratified cross-validation (i.e., 60%/20%/20% training/validation/test split). The performance is evaluated using the area under the precision-recall curve (AUCPRC)[4], which is an unbiased and more comprehensive metric for class-imbalanced tasks compared to other metrics such as F-score, ROC, and accuracy [9].

**Comparison with Resampling Imbalanced Learning (IL) Methods.** We first compare MESA with resampling techniques, which have been widely used in practice for preprocessing imbalanced data [15]. We select 13 representative methods from 4 major branches of resampling-based IL, i.e, under/over/cleaning-sampling and over-sampling with cleaning-sampling post-process. We test all methods on the challenging highly-imbalanced (IR=111, 87,450 samples) *Protein Homo.* task to check their efficiency and effectiveness. Five different classifiers, i.e., K-nearest neighbor (KNN), Gaussian Naïve Bayes (GNB), decision tree (DT), adaptive boosting (Boost), and gradient boosting machine (GBM), were used to collaborate with different resampling approaches. We also record the number of samples used for model training and the time used to perform resampling.

Table 2 details the experiment results. We show that by learning an adaptive resampling strategy, MESA outperforms other traditional data resampling methods by a large margin while only using a small number of training instances. In such a highly imbalanced dataset, the minority class is poorly represented and lacks a clear structure. Thus over-sampling methods that rely on relations between minority objects (like SMOTE) may deteriorate the classification performance, even though they generate and use a huge number of synthetic samples for training. On the other hand, under-sampling methods drop most of the samples according to their rules and results in significant information loss and poor performance. Cleaning-sampling methods aim to remove noise from the dataset, but the resampling time is considerably high and the improvement is trivial.

**Comparison with Ensemble Imbalanced Learning (EIL) Methods.** We further compare MESA with 7 representative EIL methods on four real-world imbalanced classification tasks. The baselines include 4 under-sampling-based EIL methods, i.e., RUSBOOST [35], UNDERBAGGING [2], SPE [31], CASCADE [29], and 3 over-sampling-based EIL methods, i.e., SMOTEBOOST [7], SMOTEBAGGING [42] and RAMOBOOST [8]. We use the decision tree as the base learner for all EIL methods following the settings of most of the previous works [15].

We report the AUCPRC score of various under-sampling-based EIL methods with different ensemble sizes ($k$=5, 10, 20) in Table 3. The results show that MESA achieves competitive performance on various real-world tasks. For the baseline methods, we can observe that RUSBOOST and UN-

Table 3: Comparisons of MESA with other representative under-sampling-based EIL methods.

| Method | Optical Digits (IR=9.1) | | | Spectrometer (IR=11) | | | ISOLET (IR=12) | | | Mammography (IR=42) | | |
|---|---|---|---|---|---|---|---|---|---|---|---|---|
| | $k$=5 | $k$=10 | $k$=20 | $k$=5 | $k$=10 | $k$=20 | $k$=5 | $k$=10 | $k$=20 | $k$=5 | $k$=10 | $k$=20 |
| RUSBOOST [35] | 0.883 | 0.946 | 0.958 | 0.686 | 0.784 | 0.786 | 0.696 | 0.770 | 0.789 | 0.348 | 0.511 | 0.588 |
| UNDERBAGGING [2] | 0.876 | 0.927 | 0.954 | 0.610 | 0.689 | 0.743 | 0.688 | 0.768 | 0.812 | 0.307 | 0.401 | 0.483 |
| SPE [31] | 0.906 | 0.959 | 0.969 | 0.688 | 0.777 | 0.803 | 0.755 | 0.841 | 0.895 | 0.413 | 0.559 | 0.664 |
| CASCADE [29] | 0.862 | 0.932 | 0.958 | 0.599 | 0.754 | 0.789 | 0.684 | 0.819 | 0.891 | 0.404 | 0.575 | 0.670 |
| MESA (OURS) | **0.929** | **0.968** | **0.980** | **0.723** | **0.803** | **0.845** | **0.787** | **0.877** | **0.921** | **0.515** | **0.644** | **0.705** |

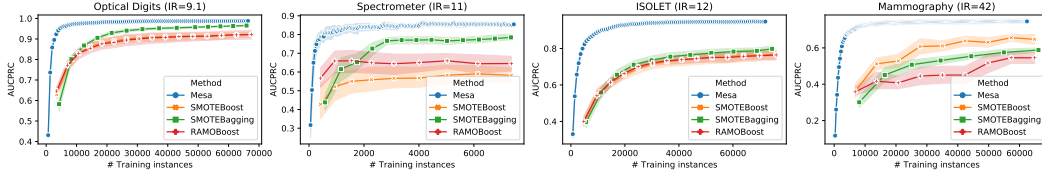

Figure 4: Comparisons of MESA with other representative over-sampling-based EIL methods.

DERBAGGING suffer from information loss as random under-sampling may discard samples with important information, and such effect is more apparent on highly imbalanced task. In comparison, the improved sampling strategies of SPE and CASCADE enable them to achieve relatively better performance but still underperform MESA. Moreover, as MESA provides an adaptive resampler that makes the ensemble training converge faster and better, its advantage is particularly evident when using small ensemble in the highly-imbalanced task. On the *Mammography* dataset (IR=42), compared with the second-best score, MESA achieved 24.70%/12.00%/5.22% performance gain when $k$=5/10/20, respectively.

We further compare MESA with 3 over-sampling-based EIL (OSB-EIL) methods. As summarized in Table 1, over-sampling-based methods typically use much more (1-2×IR times) data to train each base learner than their under-sampling-based competitors, including MESA. Thus it is unfair to directly compare MESA with over-sampling-based baselines with the same ensemble size. Therefore, we plot the performance curve with regard to the number of instances used in ensemble training, as shown in Fig. 4.

It can be observed that our method MESA consistently outperforms over-sampling-based methods, especially on highly imbalanced/high-dimensional tasks (e.g., ISOLET with 617 features, Mammo. with IR=42). MESA also shows high sample efficiency and faster convergence speed. Compared with the baselines, it only requires a few training instances to converge to a strong ensemble classifier. MESA also has a more stable training process. The OSB-EIL methods perform resampling by analyzing and reinforcing the structure of minority class data. When the dataset is small or highly-imbalanced, the minority class is usually under-represented and lacks a clear structure. The performance of these OSB-EIL methods thus becomes unstable under such circumstances.

**Cross-task Transferability of the Meta-sampler.** One important feature of MESA is its cross-task transferability. As the meta-sampler is trained on task-agnostic meta-data, it is *not* task-bounded and is directly applicable to new tasks. This provides MESA with better scalability as one can directly use a pre-trained meta-sampler in new tasks thus greatly reduce the meta-training cost. To validate this, we use *Mammography* and *Protein Homo.* as two larger and highly-imbalanced meta-test tasks, then consider five meta-training tasks including the original task (baseline), two sub-tasks with 50%/10% of the original training set, and two small tasks *Optical Digits* and *Spectrometer*.

Table 4 reports the detailed results. We can observe that the transferred meta-samplers generalize well on meta-test tasks. Scaling down the number of meta-training instances has a minor effect on the obtained meta-sampler, especially when the original task has a sufficient number of training samples (e.g., for *Protein Homo.*, reducing the meta-training set to 10% subset only results in -0.10%/-0.34% Δ when $k$=10/20). Moreover, the meta-sampler that trained on a small task also demonstrates noticeably satisfactory performance (superior to other baselines) on new, larger, and even heterogeneous tasks, which validates the generality of the proposed MESA framework. Please refer to the Appendix for a comprehensive cross/sub-task transferability test and other additional experimental results.

Table 4: Cross-task transferability of the meta-sampler.

| Meta-test Meta-train | Mammography (IR=42, 11,183 instances) | | | | Protein Homo. (IR=111, 145,751 instances) | | | |
|---|---|---|---|---|---|---|---|---|
| | $k$=10 | $\Delta$ | $k$=20 | $\Delta$ | $k$=10 | $\Delta$ | $k$=20 | $\Delta$ |
| 100% | 0.644±0.028 | baseline | 0.705±0.015 | baseline | 0.840±0.009 | baseline | 0.874±0.008 | baseline |
| 50% subset | 0.642±0.032 | -0.30% | 0.702±0.017 | -0.43% | 0.839±0.009 | -0.12% | 0.872±0.009 | -0.23% |
| 10% subset | 0.640±0.031 | -0.62% | 0.700±0.017 | -0.71% | 0.839±0.008 | -0.10% | 0.871±0.006 | -0.34% |
| Optical Digits | 0.637±0.029 | -1.09% | 0.701±0.015 | -0.57% | 0.839±0.006 | -0.12% | 0.870±0.006 | -0.46% |
| Spectrometer | 0.641±0.025 | -0.54% | 0.697±0.021 | -1.13% | 0.836±0.009 | -0.48% | 0.870±0.006 | -0.46% |

## 5  Conclusion

We propose a novel imbalanced learning framework MESA. It contains a meta-sampler that adaptively selects training data to learn effective cascade ensemble classifiers from imbalanced data. Rather than following random heuristics, MESA directly optimizes its sampling strategy for better generalization performance. Compared with prevailing meta-learning IL solutions that are limited to be co-optimized with DNNs, MESA is a generic framework capable of working with various learning models. Our meta-sampler is trained over task-agnostic meta-data and thus can be transferred to new tasks, which greatly reduces the meta-training cost. Empirical results show that MESA achieves superior performance on various tasks with high sample efficiency. In future work, we plan to explore the potential of meta-knowledge-driven ensemble learning in the long-tail multi-classification problem.

## 6  Statement of the Potential Broader Impact

In this work, we study the problem of *imbalanced learning* (IL), which is a common problem related to machine learning and data mining. Such a problem widely exists in many real-world application domains such as finance, security, biomedical engineering, industrial manufacturing, and information technology [15]. IL methods, including the proposed MESA framework in this paper, aim to fix the bias of learning models introduced by skewed training class distribution. We believe that proper usage of these techniques will lead us to a better society. For example, better IL techniques can detect phishing websites/fraud transactions to protect people's property, and help doctors diagnose rare diseases/develop new medicines to save people's lives. With that being said, we are also aware that using these techniques improperly can cause negative impacts, as misclassification is inevitable in most of the learning systems. In particular, we note that when deploying IL systems in medical-related domains, misclassification (e.g., failure to identify a patient) could lead to medical malpractice. In such domains, these techniques should be used as auxiliary systems, e.g., when performing diagnosis, we can adjust the classification threshold to achieve higher recall and use the predicted probability as a reference for the doctor's diagnosis. While there are some risks with IL research, as we mentioned above, we believe that with proper usage and monitoring, the negative impact of misclassification could be minimized and IL techniques can help people live a better life.

### Acknowledgments and Disclosure of Funding

We thank anonymous referees for their constructive suggestions on improving the paper. This work is supported by the National Natural Science Foundation of China (No.61976102, No.U19A2065).

## Footnotes

[2]To avoid confusion, in Eq. 1, we use $|\cdot|$ and $abs(\cdot)$ to denote cardinality and absolute value, respectively.

[3]Imbalance ratio (IR) is defined as $|\mathcal{N}|/|\mathcal{P}|$.

[4]All results are averaged over 10 independent runs.

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
