[Supplementary Material]

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

[5] https://imbalanced-learn.readthedocs.io/en/stable/api.html

[6] https://github.com/dialnd/imbalanced-algorithms

[7] https://github.com/ZhiningLiu1998/self-paced-ensemble

[8] `https://github.com/ZhiningLiu1998/mesa`

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

# A  Additional Results

## A.1  Cross-task and sub-task transferability of the meta-sampler

(a) Cross-task transfer performance.

(b) Cross-task transfer performance loss.

(c) Sub-task transfer performance.

(d) Sub-task transfer performance loss.

Figure 5: Cross/Sub-task transfer performance loss of MESA.

In addition to results reported in Table 4, we conduct further experiments on all five tasks to test the cross-task transferability of the meta-sampler. The results are presented in Fig. 5 (with $k$=10). For the cross-task transfer experiment, we meta-train the meta-sampler on each task separately, then apply it on other unseen meta-test tasks. As shown in Fig. 5(b), in all 20 heterogenous training-test task pairs, 18/20 of them manage to have less than 1% performance loss. On the other hand, in the sub-task transfer experiment, for each task, we meta-train the meta-sampler on 100%/50%/25%/10%/5% subset, then apply it back to the original full dataset. Again MESA shows robust performance, in all 20 subset transfer experiments, 17/20 of them manage to have less than 1% performance loss. The effect of reducing the meta-training set scale is more significant in small datasets. The largest performance loss (-1.64%) is reported in {5%, *Spectrometer*} setting, which is the smallest dataset with only 531 instances. For large datasets, scaling down the meta-training set greatly reduces the number of instances as well as meta-training costs, while only brought about minor performance loss, e.g., -0.23% loss in {5%, *Protein Homo.*}.

## A.2  Robustness to corrupted labels.

In practice, the collected training dataset may contain corrupted labels. Typical examples include data labeled by crowdsourcing systems or search engines [20, 29]. The negative impact brought by noise is particularly prominent on skewed datasets that inherently have an unclear minority data structure. In this experiment, *Mammography* and *Protein Homo.* tasks are used to test the robustness of different EIL methods on highly-imbalanced datasets. We simulate real-world corrupted labels by introducing flip noise. Specifically, flip the labels of $r_{\text{noise}}\%$ (i.e., $|\mathcal{P}| \cdot r_{\text{noise}}$) minority samples in the

Table 5: Generalized performances on real-world imbalanced datasets with varying label noise ratios.

| Dataset | Mammography (IR=42, 11,183 instances) | | | | Protein Homo. (IR=111, 145,751 instances) | | | |
|---|---|---|---|---|---|---|---|---|
| Method | $r_{noise}$=0% | $r_{noise}$=10% | $r_{noise}$=25% | $r_{noise}$=40% | $r_{noise}$=0% | $r_{noise}$=10% | $r_{noise}$=25% | $r_{noise}$=40% |
| RUSBOOST [39] | 0.511 | 0.448 | 0.435 | 0.374 | 0.738 | 0.691 | 0.628 | 0.502 |
| UNDERBAGGING [2] | 0.401 | 0.401 | 0.375 | 0.324 | 0.632 | 0.629 | 0.629 | 0.617 |
| SPE [34] | 0.559 | 0.476 | 0.405 | 0.345 | 0.819 | 0.775 | 0.688 | 0.580 |
| CASCADE [32] | 0.575 | 0.540 | 0.447 | 0.357 | 0.805 | 0.781 | 0.708 | 0.594 |
| MESA (OURS) | **0.644** | **0.618** | **0.493** | **0.401** | **0.840** | **0.806** | **0.757** | **0.677** |

(a) Performance in meta-training tasks      (b) Performance in meta-test tasks

Figure 6: Visualization of MESA's cross-task meta-training process (slide mean window = 50).

training set from 1 to 0. Accordingly, an equal number of majority samples are flipped from 0 to 1. We thereby get a noisy dataset with the same IR. For each dataset, we test the USB-EIL methods with $k = 10$ trained on the 0%/10%/25%/40% noisy training sets.

The results are summarized in Table 5, which shows that MESA consistently outperforms other baselines under different levels of label noise. The meta-sampler $\Im$ in MESA can efficiently prevent the ensemble classifier from overfitting noise as it is optimized for generalized performance, while the performance of other methods decrease rapidly as the noise level increases. Compared with the second-best baselines, MESA achieves 12.00%/14.44%/10.29%/7.22% (*Mammography*) and 2.56%/3.20%/6.92%/9.72% (*Protein Homo.*) performance gain when $r_{noise}$=0%/10%/25%/40%.

## A.3    Cross-task meta-training

In the meta-training process of MESA, collecting transitions is independent of the updates of the meta-sampler. This enables us to simultaneously collect meta-data from multiple datasets and thus to co-optimize the meta-sampler over these tasks. There may be some states that can rarely be observed in a specific dataset, in such case, parallelly collecting transitions from multiple datasets also helps our meta-sampler exploring the state space and learns a better policy. Moreover, as previously discussed, a converged meta-sampler can be directly applied to new and even heterogeneous tasks. Hence by cross-task meta-training, we can obtain a meta-sampler that not only works well on training tasks but is also able to boost MESA's performance on unseen (meta-test) tasks. To verify this, we follow the setup in section 4.2 using two small tasks for cross-task meta-training and two large tasks for the meta-test. We plot the generalized performance on all the four tasks during the cross-task meta-training process, as shown in Fig 6. The performance scores of other representative EIL methods from Table 3 are also included. Note that we only plot the two best performing baselines in each subfigure for better visualization.

At the very start of meta-training, the meta-sampler $\Im$ is initialized with random weights. Its performance is relatively poor at this point. But as meta-training progresses, $\Im$ adjusts its sampling strategy to maximize the expected generalized performance. After 50-60 training episodes, MESA surpasses the best performing baseline method and continued to improve. Finally, we get a meta-sampler that is able to undertake adaptive under-sampling and thereby outperform other EIL methods on all meta-training and meta-test tasks.

Table 6: Ablation study of MESA on 4 real-world datasets. Random policy refers to using randomly initialized meta-sampler to perform meta-sampling. $k$ represents the ensemble size. $\Delta$ is the relative performance loss (%) compared to MESA policy.

| Dataset | Method | $k = 5$ | $\Delta$ | $k = 10$ | $\Delta$ | $k = 20$ | $\Delta$ |
|---|---|---|---|---|---|---|---|
| Optical Digits | MESA policy | 0.929 | baseline | 0.968 | baseline | 0.980 | baseline |
| | Random policy | 0.904 | -1.61% | 0.959 | -0.93% | 0.975 | -0.51% |
| | Random sampling | 0.876 | -5.71% | 0.927 | -4.24% | 0.954 | -2.65% |
| Spectrometer | MESA policy | 0.723 | baseline | 0.803 | baseline | 0.845 | baseline |
| | Random policy | 0.685 | -5.26% | 0.774 | -3.61% | 0.800 | -5.33% |
| | Random sampling | 0.610 | -15.63% | 0.692 | -13.82% | 0.755 | -10.65% |
| ISOLET | MESA policy | 0.787 | baseline | 0.877 | baseline | 0.921 | baseline |
| | Random policy | 0.748 | -4.96% | 0.849 | -3.19% | 0.891 | -3.26% |
| | Random sampling | 0.688 | -12.58% | 0.768 | -12.43% | 0.812 | -11.83% |
| Mammography | MESA policy | 0.515 | baseline | 0.644 | baseline | 0.705 | baseline |
| | Random policy | 0.405 | -21.36% | 0.568 | -11.80% | 0.662 | -6.10% |
| | Random sampling | 0.307 | -40.39% | 0.401 | -37.73% | 0.483 | -31.49% |

Table 7: Description of the real-world imbalanced datasets.

| Dataset | Repository | Target | Imbalance Ratio | #Samples | #Features |
|---|---|---|---|---|---|
| Optical Digits | UCI | target: 8 | 9.1:1 | 5,620 | 64 |
| Spectrometer | UCI | target: $\geq 44$ | 11:1 | 531 | 93 |
| ISOLET | UCI | target: A, B | 12:1 | 7,797 | 617 |
| Mammography | UCI | target: minority | 42:1 | 11,183 | 6 |
| Protein Homo. | KDDCUP 2004 | target: minority | 111:1 | 145,751 | 74 |

## A.4 Ablation study

To assess the importance of Gaussian function weighted meta-sampling and meta-sampler respectively, we carry out ablation experiments on 4 real-world datasets. They are Optical Digits, Spectrometer, ISOLET, and Mammography with increasing IR (9.1/11/12/42). Our experiments shown in Table 6 indicate that MESA significantly improves performance, especially when using small ensembles on highly imbalanced datasets.

## B  Implementation Details

**Datasets.** All datasets used in this paper are publicly available, and are summarized in Table 7. One can fetch these datasets using the `imblearn.dataset` API[5] of the imbalanced-learn [27] Python package. For each dataset, we keep-out the 20% validation set and report the result of 4-fold stratified cross-validation (i.e., 60%/20%/20% train/valid/test split). We also perform class-wise split to ensure that the imbalanced ratio of the training, validation, and test sets after splitting is the same.

**Base classifiers.** All used base classifiers (i.e., K-nearest neighbor classifier, Gaussian native bayes, decision tree, adaptive boosting, gradient boosting machine) are implemented using `scikit-learn` [36] Python package. For the ensemble models (i.e., adaptive boosting and gradient boosting), we set the `n_estimators = 10`. All other parameters use the default setting specified by the `scikit-learn` package.

**Implementation of baseline methods.** All baseline resampling IL methods (RANDOMUS, NEARMISS [35], CLEAN [26], ENN [47], TOMEKLINK [45], ALLKNN [44], OSS [24], SMOTE [6], ADASYN [17], BORDERSMOTE [16], SMOTEENN [4], and SMOTETOMEK [3]) are implemented in `imbalanced-learn` Python package [27]. We directly use their implementation and default hyper-parameters in our experiments. We use open-source code[67] for implementation of

Table 8: Hyper-parameters of EIL baselines.

| Method | Hyper-parameter | Value |
|---|---|---|
| RUSBOOST [39] | n_samples | 100 |
| | min_ratio | 1.0 |
| | with_replacement | True |
| | learning_rate | 1.0 |
| | algorithm | SAMME.R |
| SMOTEBOOST [7] | n_samples | 100 |
| | k_neighbors | 5 |
| | learning_rate | 1.0 |
| | algorithm | SAMME.R |
| RAMOBOOST [8] | n_samples | 100 |
| | k_neighbors_1 | 5 |
| | k_neighbors_2 | 5 |
| | alpha | 0.3 |
| | learning_rate | 1.0 |
| | algorithm | SAMME.R |
| UNDERBAGGING [2] | - - - | - - - |
| SMOTEBAGGING [46] | k_neighbors | 5 |
| BALANCECASCADE [32] | - - - | - - - |
| SELFPACEDENSEMBLE [34] | hardness_func | cross entropy |
| | k_bins | 10 |

Table 9: Hyper-parameters of SAC [14].

| Hyper-parameter | Value |
|---|---|
| Policy type | Gaussian |
| Reward discount factor ($\gamma$) | 0.99 |
| Smoothing coefficient ($\tau$) | 0.01 |
| Temperature parameter ($\alpha$) | 0.1 |
| Learning rate | 1e-3 |
| Learning rate decay steps | 10 |
| Learning rate decay ratio | 0.99 |
| Mini-batch size | 64 |
| Replay memory size | 1e3 |
| Steps of gradient updates | 1e3 |
| Steps of random actions | 5e2 |

Table 10: Hyper-parameters of MESA.

| Hyper-parameter | Value |
|---|---|
| Meta-state size | 10 |
| Gaussian function parameter $\sigma$ | 0.2 |

Table 11: Performance of different policy network architectures.

| Network Architecture | Optical Digits Task | | |
|---|---|---|---|
| | $k$=5 | $k$=10 | $k$=20 |
| {10, 50, 1} | 0.929±0.015 | 0.968±0.007 | 0.980±0.003 |
| {10, 100, 1} | 0.930±0.014 | 0.966±0.007 | 0.979±0.004 |
| {10, 200, 1} | 0.922±0.018 | 0.964±0.008 | 0.978±0.005 |
| {10, 25, 25, 1} | 0.928±0.014 | 0.966±0.007 | 0.980±0.004 |
| {10, 50, 50, 1} | 0.929±0.017 | 0.967±0.008 | 0.978±0.004 |
| {10, 100, 100, 1} | 0.926±0.015 | 0.966±0.010 | 0.979±0.006 |
| {10, 10, 10, 10, 1} | 0.924±0.013 | 0.964±0.007 | 0.977±0.004 |
| {10, 25, 25, 25, 1} | 0.924±0.016 | 0.966±0.006 | 0.978±0.002 |
| {10, 50, 50, 50, 1} | 0.926±0.006 | 0.965±0.006 | 0.979±0.005 |

baseline ensemble imbalanced learning (EIL) methods (RUSBOOST [39], UNDERBAGGING [2], CASCADE [32], SPE [34], SMOTEBOOST [7], SMOTEBAGGING [46], and RAMOBOOST [8]). The hyper-parameters of these baseline EIL methods are reported in Table 8.

**Implementation of** MESA. MESA is implemented with `PyTorch`. The empirical results reported in the paper use hyper-parameters in Tables 9 and 10 for the meta-training of MESA. We open-sourced our MESA implementation at Github[8] with a *jupyter notebook* file that allows you to quickly (I) conduct a comparative experiment, (II) visualize the meta-training process of MESA, and (III) visualize the experimental results. Please check the repository for more information.

The actor policy network of meta-sampler is a multi-layer perceptron with one hidden layer containing 50 nodes. Its architecture is thus {`state_size`, 50, 1}. The corresponding (target) critic Q-network is also an MLP but with two hidden layers. As it takes both state and action as input, its architecture is thus {`state_size+1`, 50, 50, 1}. Each hidden node is with ReLU activation function, and the output of the policy network is with the tanh activation function, to guarantee the output located in the interval of [0, 1]. As a general network training trick, we employ the Adam optimizer to optimize the policy and critic networks.

We test different network architecture settings in experiments. Table 11 depicts some representative results under 9 different policy network structures, with different depths and widths. It can be observed that varying MLP settings have no substantial effects on the final result. We hence prefer to use the simple and shallow one.

(a) Sub-task transfer performance.　　　(b) Sub-task meta-training time.

Figure 7: The influence of scaling down the meta-training set.

## C Discussion

### C.1 Complexity analysis of the proposed framework

Our MESA framework can be roughly regarded as an under-sampling-based ensemble imbalanced learning (EIL) framework (Algorithm 2) with an additional sampler meta-training process (Algorithm 3).

**Ensemble training.** Given an imbalanced dataset $\mathcal{D}$ with majority set $\mathcal{N}$ and minority set $\mathcal{P}$, where $|\mathcal{N}| \gg |\mathcal{P}|$. Suppose that the cost of training a base classifier $f(\cdot)$ with $N$ training instances is $C_{f\text{train}}(N)$. As MESA performs strictly balanced under-sampling to train each classifier, we have

$$\text{Cost of } k\text{-classifier ensemble training} : k \cdot C_{f\text{train}}(2|\mathcal{P}|)$$

In comparison, the cost is $k \cdot C_{f\text{train}}(|\mathcal{N}| + |\mathcal{P}|)$ for reweighting-based EIL methods (e.g., ADABOOST) and around $k \cdot C_{f\text{train}}(2|\mathcal{N}|)$ for over-sampling-based EIL methods (e.g., SMOTEBAGGING).

**Meta-training.** Let's denote the cost of performing a single gradient update step of the meta-sampler $\Im$ as $C_{\Im\text{update}}$, this cost mainly depends on the choice of the policy/critic network architecture. It is barely influenced by other factors such as the dataset size in ensemble training. In our MESA implementation, we do $n_{\text{random}}$ steps for collecting transitions with random actions before start updating $\Im$, and $n_{\text{update}}$ steps for collecting online transitions and perform gradient updates to $\Im$. Then we have

$$\text{Cost of meta-training} : (n_{\text{random}} + n_{\text{update}}) \cdot C_{f\text{train}}(2|\mathcal{P}|) + n_{\text{update}} \cdot C_{\Im\text{update}}$$

As mentioned before, the meta-training cost can be effectively reduced by scaling down the meta-training dataset (i.e., reducing $|\mathcal{P}|$). This can be achieved by using a subset of the original data in meta-training. One can also directly use a meta-sampler pre-trained on other (smaller) dataset to avoid the meta-training phase when applying MESA to new tasks. Both ways should only bring minor performance loss, as reported in Fig. 5.

Note that, reducing the number of meta-training instances only influences the $C_{f\text{train}}(\cdot)$ term. Therefore, the larger the $C_{f\text{train}}(2|\mathcal{P}|)/C_{\Im\text{update}}$, the higher the acceleration ratio brought by shrinking the meta-training set. We also show some results in Fig. 7 to demonstrate such influence. The decision tree classifier we used has no max depth limitation, thus its training cost is higher when dealing with high-dimensional data. We thus choose three tasks with different numbers of features for the test, they are *Mammography/Protein Homo./ISOLET* with 6/74/617 features. It can be observed that the acceleration effect is slightly weaker for the low-dimensional *Mammography* task, as the cost of training base classifier is small compared with the cost of updating meta-sampler. On the other hand, for those high-dimensional tasks (i.e., *ISOLET* and *Protein Homo.*), shrinking the meta-training set greatly reduces the cost of meta-sampler training as we expect.

Figure 8: Learned meta-sampler policies on *Mammography* dataset with varying label noise ratios.

## C.2 Guideline of selecting MESA hyper-parameters

The meta-state size $b$ determines how detailed our error distribution approximation is (i.e., the number of bins in the histogram). Thus setting a small meta-state size may lead to poor performance. Increasing it to a large value (e.g., $\geq 20$) brings greater computational cost but only trivial performance increment. We recommend setting the meta state size to be 10. One can try a bigger meta-state when working on larger datasets.

The Gaussian function parameter $\sigma$ determines how to execute meta-sampling in MESA. Specifically, given an action $\mu$, we expect the meta-sampling selects those instances with error values close to $\mu$. Besides, the meta-sampling is also responsible for providing diversity, which is an important characteristic in classifiers combination. A small $\sigma$ can guarantee to select examples with small errors around $\mu$, but this would result in subsets that lack diversity. For example, in the late iterations of ensemble training, most of the data instances have stable error values, and meta-sampling with small $\sigma$ will always return the same training set for a specific $\mu$. This is detrimental to further improve the ensemble classifier. Setting a large $\sigma$ will "flatten " the Gaussian function, more instances with different errors are likely to be selected and thus bring more diversity. However, when $\sigma \rightarrow \infty$, the meta-sampling turns into uniform random under-sampling that makes no sense for meta-training. We also note that although one can expand the policy to automatically determine $\sigma$, it requires additional computational cost and the benefit is very limited. More importantly, selecting inappropriate $\sigma$ will interfere with the quality of collected transitions, causing an unstable meta-training process. Therefore, we suggest using $\sigma = 0.2$ to balance between these factors.

## D   Visualization

### D.1   Visualization of learned meta-sampler policy

We visualize the learned meta-sampler policy under different levels of noises in Fig. 8. It clearly shows that the sampling strategy becomes more conservative as the noise ratio grows. At the very start of ensemble training, there are only a few base learners and thus the ensemble classifier underfits the training set. At this point, the meta-sampler tends to select training instances with larger errors, hence accelerating the fitting process. It continues to use such a strategy on datasets with no/few noises. However, on highly noisy datasets (e.g., with label noise ratio $\geq 40\%$), the meta-sampler prefers to select training instances with relatively lower errors in later iterations as the hard-to-classify instances are likely to be noises/outliers. This effectively prevents the ensemble classifier from overfitting noisy data points.

### D.2   Visualization of meta-training process

We visualize the meta-training process in Fig. 9. As the meta-training progress, the classification performance shows consistent improvement in training, validation, and test set in all tasks.

Figure 9: Train/Validation/Test performance during meta-training process (slide mean window=50).