[Reviews · NeurIPS 2020]

Review 1

Summary and Contributions: The paper introduces a so-called meta sampler that adaptivity undersamples imbalanced data, using an iterative approach. The authors compare their work to others working in the field and the results indicate that their method is promising.

Strengths: This is an important topic of research.

Weaknesses: The algorithm focuses on undersampling, rather than oversampling. This is worrisome, since the comparative study is with SMOTE-like algorithms. Thus, one would expect to also see some comparison with random oversampling, etc. and a deeper explanation about this design choice. The paper is difficult to follow and needs to be edited prior to publication.

Correctness: n equation 3, authors decide to use the Gaussian function for determining the sampling weight. This choice needs some justification, and also the fact that these weights are unnormalised needs to be verified. Further, the choice of using the "soft actor critic" approach is not motivated. The results in Tables 2 and 3 come as a surprise. Notably, the poor performance of the SMOTE-based methods seems suspect and needs a careful explanation.

Clarity: The clarity could be improved, notably when explaining the design decisions and the experimental results. The paper contains numerous typographical errors; for instance the authors refer to "Native Bayes"? Also, the ordering of the references are incorrect, e.g. [32, 25, 6] instead of [6, 25, 32].

Relation to Prior Work: The authors missed many related works, notable in the cost-sensitive learning domain.

Reproducibility: No

Additional Feedback:


Review 2

Summary and Contributions: Imbalanced learning has been a hot topic in machine learning since it can be easily encountered in most real-world applications, where data are not only scarce but also heavily biased toward one class. Mesa, the proposed framework, aims at tackling this learning problem through a novel approach combining meta-learning, resampling techniques, ensemble learning and reinforcement learning. While theoretical justifications are meagre, the paper is rich in technical details on how the 4 aforementioned frameworks are combined, as well as empirical results demonstrating the effectiveness of Mesa.

Strengths: The main strength of the paper resides in combining 4 established learning frameworks into a new (and novel) learning approach. The resulting algorithm is made up of three separate procedures: resampling, ensemble learning and meta learning. In my opinion, the most important contribution of this paper is the cross-task transferability of the meta-sampler, which opens up several interesting research directions, e.g. applying a pre-trained meta-sampler to small datasets (where classical IL methods usually fail to train pertinent models) and large datasets (scalability). The empirical section is also a strong contribution as it is particularly rich in results covering various facets of Mesa.

Weaknesses: I have only a few remarks on this paper, even though they shouldn't be considered as weaknesses. They are listed below in no particular order. - in eq.1 | is used both as the absolute value operator and the cardinality one, which can lead to confusion - in eq.2, \tau and v have not been previously defined (unless I'm missing something) - I find it regrettable that no theoretical analysis of Mesa (e.g. convergence speed, generalization error, etc) is proposed aside from the complexity one, especially since it is built upon frameworks with strong theoretical properties - line 155 "is thus can be" typo - line 173 reference error "Haarnoja et al." - line 233 compares > compared - table 2, what does k correspond to? Is it the parameter of Algorithm 2? - a few more datasets would've been appreciated, especially concerning the cross-task transferability

Correctness: There are little to no theoretical results in this paper, as it is clearly a more empirical one. The experimentation protocol is clearly explained and defined.

Clarity: The paper is very well written and easy to follow. Quite a pleasant read.

Relation to Prior Work: The authors propose an extensive review of related work and do a nice job at stating the importance of the contribution w.r.t. existing works.

Reproducibility: Yes

Additional Feedback: after rebuttal: I find the answers provided in the rebuttal satisfying, as such I'm keeping my original rating for this paper.


Review 3

Summary and Contributions: This paper targets on addressing the binary imbalance classification problem, where the negative samples are much more than the positive samples. The main idea is to learn an ensemble of classifiers, each of which is learned with instances sampled by a meta-sampler. The meta-sampler takes input the histogram error distribution of training and validation set, and outputs a scalar value \mu, from which the sampling weight is calculated. The instances with error closer to \mu have higher weights for sampling. For mapping the histogram error distribution to value \mu, authors proposed to learn a policy network.

Strengths: The designed meta-sampler can adaptively sample the instances to construct a balanced positive and negative training dataset, from which a member of the classifier ensemble is learned. Therefore, the proposed approach performs well on the imbalanced classification problem, even with noise and corrupted labels. Overall, it is an interesting work. The meta-sampler can effectively address the class imbalance problem.

Weaknesses: There are some errors in Algorithm 1, which may be typos. These errors are given in detailed comments later.

Correctness: The proposed solution has been compared with several groups of baselines. The evaluation results show the superior performance of the proposed solution. In addition, extensive results in the supplementary document further confirms the usefulness of the proposed meta-sampler.

Clarity: The paper is well written.

Relation to Prior Work: The related work is clearly discussed, and the contribution is appropriately highlighted.

Reproducibility: Yes

Additional Feedback: In algorithm 1, line 1, P_t is the majority and N_t is the minority set. However, in line 3, why sampling majority subset N’_t from N_t? In each iteration, the training subset is sampled from the original training set. That’s to say, for each training instance in the original training set, a classification error is evaluated by the current ensemble model F_{t}? However, in line 2 of Algorithm 1, x_i is classified by F. And errors like "A instance is " should be corrected. Authors' feedback addressed my comments.


Review 4

Summary and Contributions: This paper deals with supervised learning (classification) under class imbalance. The authors propose to address this issue by using a meta/ensemble-learning framework. In this framework, the meta-algorithm deduces an appropriate data sampling strategy that generates a data set for a new base learner to train. The meta-learner is trained using reinforcement learning. The meta-state is composed of two histograms that are respectively the empirical distributions of the training and validation error. The meta-sampler uses this state to sample a coefficient. This coefficient is used as the mean of a Gaussian from which sampling weights are obtained. These weights are used to obtain a balanced dataset that is used for training by the new base learner. The aggregation rule of base learners is the average of their soft outputs. The validation and training errors of the updated ensemble can be computed thereby reaching a new meta-state. The meta-sampler is regarded as a policy in the RL framework, i.e. mapping states to actions. It is trained using the soft actor/critic deep RL algorithm which is suitable for continuous state and actions spaces. The main contributions are : (i) a new meta-ML framework for imbalanced data that is not tied to a specific type of base learner, (ii) a meta-learner that is task-agnostic and can be re-used for different ensembles of base learners addressing different tasks. The method compares favorably to prior arts.

Strengths: The main strength of the proposed method is its flexibility and re-usability (see reported contributions above). In my opinion, an interesting aspect w.r.t. novelty is the use of RL techniques to train the meta-learner. The paper addresses an open problem in ML and is thus perfectly suited for a submission at NeurIPS.

Weaknesses: The main weakness of the proposed framework is that a few aspects of the authors' construct are somewhat arbitrary. In particular, the choice of a Gaussian distribution to compute weights is not clearly justified and does not seem very natural. By the way, the impact of this choice is not discussed. Can't the RL agent directly issue a set of weights ? Likewise, the use of a linear aggregation rule is not much discussed. Wouldn't an RNN, i.e. a trainable non-linear aggregation that can forget about a bad base learner (such as the initial one), be better ?

Correctness: Experiments in 4.1 should be extended to several imbalance ratios to feature the ability of the proposed method to handle different imbalance schemes. This is important as the validity of the methods relies solely on experimental evidence. Also, unless I’m mistaken, the total number of training examples does not appear in these synthetic experiments. Table 2 : I am unclear with the number k of base learner used in these experiments. Error bars are only given in Figure 4 and Table 4, such information must be given on the provided results (even those with so called comfortable margin). The origin of the variance of the results should be highlighted. The impact of the train/valid/test split is not commented. The experiments should be embedded in a stratified CV loop. Finally, a comparison to other meta-ML methods would also be welcome even if they are tied to DNN framework.

Clarity: The paper reads pretty well in my opinion.

Relation to Prior Work: The increment of over prior is clearly stated and a short review of the state-of-the-art is provided.

Reproducibility: Yes

Additional Feedback: Minor comments: The error e does not appear in (3). Another notation should be used to avoid confusion with Euler’s number in the Gaussian. In alg. 1 it is clear the input of the Gaussian distribution is the error (not so clear in the text). ---- Post author response: The authors have provided a number of additional experimental details which give stronger guarantees on the quality and relevance of their results. Consequently, I have updated my score from 6 to 7 although I am still a bit unsatisfied with the justification of the Gaussian function to compute the weights.

[Author Response · NeurIPS 2020]

**We thank all reviewers for the constructive comments!** Itemized responses to each reviewer are appended below:

*Abbreviations: imbalanced learning (IL), under-sampling (US), over-sampling (OS), cost-sensitive learning (CSL).*

**To all reviewers:** Thanks for your careful reading! We will carefully resolve all writing, format, and notation issues.

**Response to reviewer #1: 1)** MESA focuses on US, which is worrisome as it was compared with SMOTE-like baselines.

**R:** We highlight that we conducted an extensive comparison including 19 different baselines (balanced/cleaning US,

distance/ranking-based OS, their ensemble variants, etc.). SMOTE-*like algorithms are only a small part of them.*

**2)** Results of random OS (ROS). **R:** We have tested ROS but didn't report the results due to the page limit. Generally, it

yields poor performance compared with the other baselines. These results will be included in the camera-ready version.

**3)** About the use of Gaussian function. **R:** Our main goal is to design an *efficient*, *concise*, and *practical* IL framework.

It is nearly impossible to make instance-level decisions by using a complex meta-sampler (e.g., set a large output layer

or use RNN), as the complexity of a single update will grow from $\mathcal{O}(1)$ to $\mathcal{O}(n)$, where the number of instances $n$ is

usually large in real-world datasets. Besides, complex model architecture also brings extra memory cost and hardship in

optimization. For these reasons, we choose to use such a Gaussian function trick to simplify the meta-sampler, while

maintains the ability to perform controlled sampling for generating accurate and diverse base learners.

**4)** About weight normalization. **R:** For clarity, Eq. 3 shows the unnormalized sampling weights (noted in the paper).

**5)** About SAC. **R:** SAC has better suitability for continuous state & action spaces compares to other methods like DQN.

**6)** Why SMOTE-like methods perform poorly? **R:** We have discussed this in the paper, please see our discussions at

lines 33-37, lines 224-227, lines 256-259. Also, please see the analysis and results in [1, 2] and references therein.

**7)** About related works. **R:** We conducted an extensive review of related works including 3 papers of CSL (lines 78-82).

As MESA is US-based, we mainly focused on more closely related works rather than reweighting ones, such as CSL.

**8)** Reproducibility **R:** We have elaborated on all the implementation details in the appendix due to the page limit. The

code of this work was released via GitHub, the link can be found in the paper (footnote #1).

**Response to reviewer #2:**

**1)** Theoretical analysis. **R:** Thanks for the advice! Most of the existing theoretical results of ensemble learning are

limited to a specific model (e.g., perceptron/SVM) and ensemble schema (e.g., boosting/bagging). As MESA is a very

general framework, it is not easy to derive *useful* theoretical results, so instead, we show our insights by providing solid

experimental results and analysis. We will list the theoretical analysis of MESA as an important future direction.

**2)** More datasets. **R:** Due to both the resource and space limitations, we have tried our best to fit all important results in

the paper. We will include more datasets in the future, the results will possibly be released online (via Github).

**Response to reviewer #3:**

**1)** About algorithm 1. **R:** Sorry for the confusion! Please note that $\mathcal{P}_\tau$ and $\mathcal{N}_\tau$ stands for minority and majority set

respectively (not the other way around). The algorithm 1 actually defines a function: $\texttt{Sample}(\mathcal{D}_\tau; F, \mu, \sigma)$, where $F$

represents an ensemble classifier that is one of the function inputs. This function was called in algorithm 2, line 7, and

you can see that the corresponding input is $F_t$. We will carefully revise this part to make it more readable.

**Response to reviewer #4:**

**1)** Algorithm design. **R:** Sorry for the confusion. Please refer to our response to reviewer #1, item 3. Due to the space

limitation, we end up removed lots of discussions about out motivation. They will be included in the final version.

**2)** About ggregation rule. **R:** Thanks for your advice! Our work mainly focuses on sampling strategy, which is the core

application of meta-sampler. Adaptive aggregation, also known as "dynamic ensemble selection", is another interesting

big topic, which may be a promising extension of this paper. We will study more on this topic in future work.

**3)** About experiment 4.1. **R:** According to previous works (please see Fig. 2 in [2] and references therein) and our

experiment, we found that class-overlapping is a more important property of an IL task compares to the imbalance ratio.

Therefore, we choose to simulate different levels of overlapping for more illustrative results and visualization.

**4)** About synthetic dataset. **R:** Number of instances: $|\mathcal{P}|$=200, $|\mathcal{N}|$=2,000. We will clarify this in the final version.

**5)** About $k$ in table 2. **R:** Baselines are *resampling methods* with a single classifier trained on the resampled data ($k$=1).

**6)** About error bars. **R:** Due to page limit, we can only end up with removing the error bar from some results to fit the

8-page manuscript. We will adjust the layout to guarantee all the important results come with an error bar.

**7)** Data split. **R:** Sorry for the confusion! In our implementation, we keep-out the validation set and report the mean

score of 4-fold stratified CV (i.e., 60%/20%/20% train/valid/test split). We will clarify these in the final version.

**8)** Origin of variance. **R:** The variance comes from the randomness in the data sampling and model training process.

**9)** Comparison to other meta-ML methods. **R:** Thanks for the advice. However, due to the huge difference be-

tween them in the aspect of task (tabular/image), model (traditional classifier/neural network), and learning schema

(ensemble/standalone), it is hard to conduct a fair comparative experiment. We will depict the difference more clearly.

**References: [1]** Krawczyk B. Learning from imbalanced data: open challenges and future directions[J]. Progress in

Artificial Intelligence, 2016, 5(4): 221-232. **[2]** Liu Z, Cao W, Gao Z, et al. Self-paced Ensemble for Highly Imbalanced

Massive Data Classification[C], 2020 IEEE 36th International Conference on Data Engineering. IEEE, 2020: 841-852.


[Meta-Review · NeurIPS 2020]

Three referees support accept and one indicates reject. Following the author rebuttal the discussion was limited to two accept reviewers who both indicated that the paper is borderline (unsure if the paper was good enough, would not mind if rejected). This seems to indicate more of a score of 6 than the 7 they actually gave. However, I favour accepting the paper because it is an important research area and few general purpose ensemble techniques exist to tackle the problem. One could therefore imagine further work building on this.